# Microbiota regulates visceral pain in the mouse

**Pauline Luczynski[1†‡], Monica Tramullas[1†§], Maria Viola[1], Fergus Shanahan[1], Gerard Clarke[1,2], Siobhain O'Mahony[1,3], Timothy G Dinan[1,2], John F Cryan[1,3*]**

[1]APC Microbiome Institute, University College Cork, Cork, Ireland; [2]Department of Psychiatry and Neurobehavioural Science, University College Cork, Cork, Ireland; [3]Department of Anatomy and Neuroscience, University College Cork, Cork, Ireland

**Abstract** The perception of visceral pain is a complex process involving the spinal cord and higher order brain structures. Increasing evidence implicates the gut microbiota as a key regulator of brain and behavior, yet it remains to be determined if gut bacteria play a role in visceral sensitivity. We used germ-free mice (GF) to assess visceral sensitivity, spinal cord gene expression and pain-related brain structures. GF mice displayed visceral hypersensitivity accompanied by increases in Toll-like receptor and cytokine gene expression in the spinal cord, which were normalized by postnatal colonization with microbiota from conventionally colonized (CC). In GF mice, the volumes of the anterior cingulate cortex (ACC) and periaqueductal grey, areas involved in pain processing, were decreased and enlarged, respectively, and dendritic changes in the ACC were evident. These findings indicate that the gut microbiota is required for the normal visceral pain sensation.

**\*For correspondence:** j.cryan@ ucc.ie

[†]These authors contributed equally to this work

**Present address:** [‡]MD Program, Faculty of Medicine, University of British Columbia, Vancouver, Canada; [§]Department of Physiology and Pharmacology, University of Cantabria, Cantabria, Spain

**Competing interests:** The authors declare that no competing interests exist.

## Introduction

Accumulating evidence indicates that the gut microbiota communicates with the central nervous system (CNS) in a bidirectional manner thereby influencing brain function and behavior (*Sampson and Mazmanian, 2015*; *Dinan and Cryan, 2012*; *Mayer, 2011*). Although the majority of studies investigating the effects of the microbiota on brain function involve animal models of anxiety, depression, and cognitive dysfunction, it is now becoming clear that the gut microbiota may also have a role in other CNS-related conditions, such as visceral pain (*O'Mahony et al., 2014*; *Gareau et al., 2007*; *McKernan et al., 2010*).

Abdominal pain, often characterized by visceral hypersensitivity, is a common, and at times, dominant symptom of several gastrointestinal disorders, including functional dyspepsia and irritable bowel syndrome (IBS) (*Enck et al., 2016*). There is also a high comorbidity among visceral pain and psychiatric disorders such as depression and anxiety (*Felice et al., 2015*). These painful events are often recurring and unpredictable, which can have a debilitating impact on a person's daily life (*Quigley, 2006*). Moreover, many gastrointestinal disorders with visceral pain as a component lack an identifiable pathology and can be difficult to treat with current pharmaceuticals, many of which are associated with undesirable side effects (*Wood, 2013*; *Moloney et al., 2016*).

The perception of visceral pain is a complex process involving peripheral sensory nerves, and, in the CNS, spinal and cortical pathways as well as areas associated with integration of the experience of pain (*Apkarian et al., 2005*). Pathological pain states have been associated with altered neuroimmune signaling and glial activation in the spinal cord (*Ji et al., 2013*; *Grace et al., 2014*). In the brain, there is a significant overlap in areas regulating the affective component of visceral pain and those mediating psychological stress, a major predisposing factor for visceral hypersensitivity (*Larauche et al., 2012*). Imaging studies in humans with IBS (*Tillisch et al., 2011*; *Mertz et al.,*

**eLife digest** The human gut is home to over 100 trillion microbes collectively known as the gut microbiota. These microbes help us to digest food and absorb the nutrients effectively. A diverse and stable community of gut microbes is believed to be important for good health. Recently, it has also become clear that the microbiota can also influence the brain and how we behave. For example, many studies suggest that gut microbiota can alter how an individual perceives pain, but it is not clear how this works.

Rodents are often used in experiments as models of human biology. One of the most frequently used rodent models in studies of gut microbes is the "germ-free" mouse. These mice grow up in laboratory environments that are completely free of microbes, making it possible to study how having no gut microbes affects the health and behaviour of the mice. Luczynski, Tramullas et al. used germ-free mice to study how the gut microbiota influences an animal's sensitivity to pain.

The experiments show that, compared to mice with normal gut microbiota, the germ-free mice were more sensitive to pain from internal organs especially the gut. These mice also produced larger amounts of specific proteins involved in immune responses, which contributed to the animal's increased sensitivity to pain. Allowing the germ-free mice to be colonised with gut microbes could reverse these changes.

The experiments also show that the germ-free mice had changes in the size of two areas of the brain involved in sensing pain: an area called the anterior cingulate cortex was smaller, while the periaqueductal grey region was enlarged. There were also differences in individual nerve cells within the anterior cingulate cortex compared to normal mice.

The findings of Luczynski, Tramullas et al. reinforce the idea that the gut microbiota is involved in the sensation of pain from internal organs, and show that hypersensitivity to this form of pain can be reversed later in life by colonising the gut with microbes. Continuing to study the impact of microbes on this type of pain could aid the development of new therapies for the treatment of pain disorders in humans.

_2000_) and in animal models of visceral hypersensitivity (_Gibney et al., 2010_; _Felice et al., 2014_; _Bliss et al., 2016_) have revealed increased activation in the medial prefrontal cortex (mPFC) in response to both viscerally painful and stressful stimuli.

Numerous studies now indicate that the perception of visceral pain can by influenced by the intestinal microbiota. In rodent studies, specific bacterial strains (probiotics) have been shown to ameliorate visceral pain induced by stress (_Gareau et al., 2007_; _McKernan et al., 2010_; _Ait-Belgnaoui et al., 2006_) or antibiotic administration (_Verdú et al., 2006_), and many probiotics have been shown to benefit humans with abdominal pain (_Clarke et al., 2012_). Intriguingly, visceral hypersensitivity can be transferred via the microbiota of IBS patients to animals previously lacking microbes (_Crouzet et al., 2013_). However, the mechanisms underlying the effects of the microbiota on visceral pain perception remain to be elucidated.

Integrating these observations, this study is based on the hypothesis that the microbiota is required for the normal processing of visceral pain stimuli. To this end, we used germ-free mice (GF; mice raised without any exposure to microorganisms) to study pain-related behavior, genes, and brain structures. We first studied visceral sensitivity, expression of cytokines, Toll-like receptors (TLRs), and glial markers in the spinal cord, and mPFC morphology in these animals. Secondly, we determined if postnatal microbial colonization could normalize visceral hypersensitivity and immune status within the spinal cord of GF mice.

## Results

### Animals

Four cohorts of animals were used to perform the experiments in this paper. Animals in the first cohort (CC = 10, GF = 9; _Figure 1_) underwent CRD and then were euthanized. Animals in the second cohort (CC and GF = 9–10; _Figure 2_) were sacrificed without anesthesia and spinal cord tissue

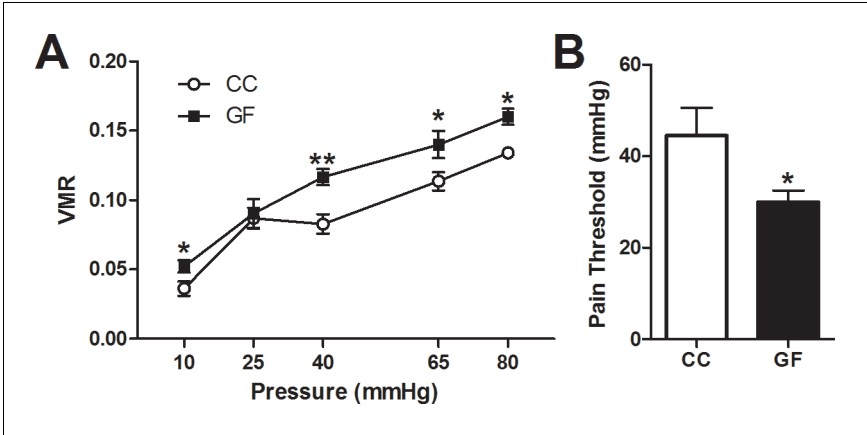

**Figure 1.** Visceral hypersensitivity in GF mice. Visceral sensitivity to colorectal distention (CRD; ascending paradigm from 10 to 80 mm Hg) was assessed as the number of visceromotor responses (VMR) over pressures. (**A**) GF mice displayed increased visceral pain responses compared to controls. (**B**) The pain threshold was lower in GF compared to CC mice. In this and subsequent figures, *p≤0.05, **p<0.01, and ***p<0.001 versus CC animals. CC, n = 10; GF, n = 9.

The following figure supplement is available for figure 1:

**Figure supplement 1.** Schematic illustration showing the ascending phasic distension paradigm (10–80 mmHg) (**A**) and representative CRD traces at the pressure of 40 mmHg and 65 mmHg for conventional (**B,C**) or germ-free (**D, E**) mice. The paradigm consists of three repeated pulses at each pressure level, with a pulse duration of 20 s at 5-min intervals.

---

was collected to perform gene expression analyses. Animals in the third cohort (CC = 20, GF = 20, GFC = 10; Figures 5, 6 and 7) underwent CRD and spinal cord tissue was collected. Animals in the fourth cohort (CC = 17, GF = 18; *Figures 3* and *4*, and *Figure 4—figure supplement 1*) were euthanized without undergoing any procedures. See Materials and methods for more details.

## Does growing up GF alter visceral sensitivity, pain-related genes, and CNS morphology?

### Visceral hypersensitivity in GF mice

When compared to controls, GF mice showed increased visceral pain responses (main effect of pressure: $F_{4,60}$ = 76.75, p<0.001; main effect of group: $F_{1,15}$ = 12.51, p=0.0030; *Figure 1A*). Post hoc analyses revealed that GF mice displayed visceral hypersensitivity at pressures of 10, 40, 65, and 80 mmHg (*Figure 1—figure supplement 1A*). Representative CRD traces at the pressure of 40 mmHg and 65 mmHg for conventional or GF mice are shown in *Figure 1—figure supplement 1B–E*. Importantly, in line with our previous studies (*O'Mahony et al., 2012*), there was no significant difference in the basal activity (between distensions) of the abdominal musculature between groups ($F_{1,15}$ = 0.38, p=0.58) or within the same experimental group ($F_{4,60}$ = 1.08, p=0.374). The visceral pain threshold was significantly lower in GF versus CC mice ($t_{17}$ = 2.12, p=0.049; *Figure 1B*).

### Altered TLR and cytokine gene expression in the spinal cord of GF mice

In GF mice, the transcription levels of TLR1 ($t_{18}$ = 2.92, p=0.0093), TLR2 ($t_{17}$ = 2.56, p=0.020), TLR3 ($t_{18}$ = 2.81, p=0.012), TLR4 ($t_{18}$ = 2.38, p=0.029), TLR5 ($t_{17}$ = 3.16, p<0.0057), TLR7 ($t_{17}$ = 2.49, p=0.023), TLR9 ($t_{16}$ = 2.22, p=0.041), and TLR12 ($t_{13}$ = 2.55, p=0.024) were significantly increased in the lumbosacral spinal cord compared to controls (*Figure 2A–H*). No changes were observed in the expression of TLR6 (CC = 0.027 ± 0.0022, GF = 0.034 ± 0.0036, $t_{18}$ = 1.54, p=0.14), TLR8 (CC = 0.018 ± 0.0021, GF = 0.023 ± 0.0022, $t_{18}$ = 1.65, p=0.12), and TLR11 (CC = 1.00 ± 0.37, GF = 0.86 ± 0.41, $t_{7}$ = 0.26, p=0.81).

In GF mice, there was also a significant increase in the transcription levels of IL6 ($t_{17}$ = 2.80, p=0.012), IL10 ($t_{17}$ = 2.11, p=0.050), TNFα ($t_{17}$ = 2.80, p=0.0015), IL1α ($t_{16}$ = 2.27, p=0.038), and

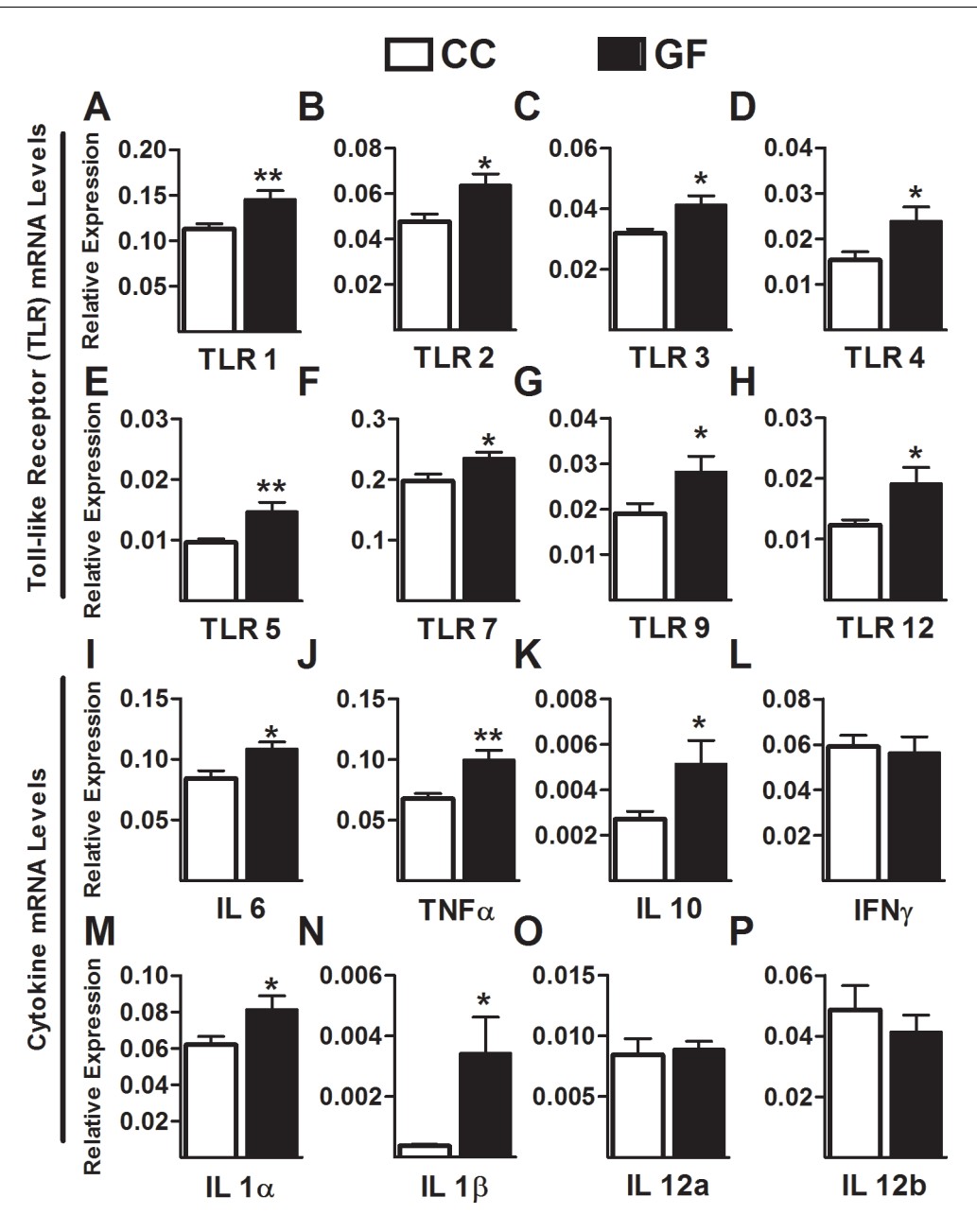

**Figure 2.** Increased Toll-like receptor and cytokine gene expression in the spinal cord of GF mice. (**A–H**) Gene expression levels of the Toll-like receptors TLR1 (**A**), TLR2 (**B**), TLR3 (**C**), TLR4 (**D**), TLR5 (**E**), TLR7 (**F**), TLR9 (**G**), and TLR12 (**H**) were significantly elevated in the spinal cord of GF versus CC mice. (**I–M**) When compared to controls, GF mice showed increased gene expression levels of the cytokines IL6 (**I**), TNFα (**J**), IL10 (**K**), IL1α (**M**), and IL1β (**N**). There was no change in the gene expression of the cytokines IFNγ (**L**), IL12α (**O**), and IL12β (**P**). CC, n = 9–10; GF, n = 9–10.

IL1$\beta$ ($t_{15}$ = 2.74, p=0.016) compared to CC mice. However, no changes were observed in the expression levels of IFN$\gamma$ ($t_{18}$ = 0.31, p=0.76), IL12a ($t_{18}$ = 0.30, p=0.77), and IL12b ($t_{18}$ = 0.75, p=0.46; *Figure 2I–P*).

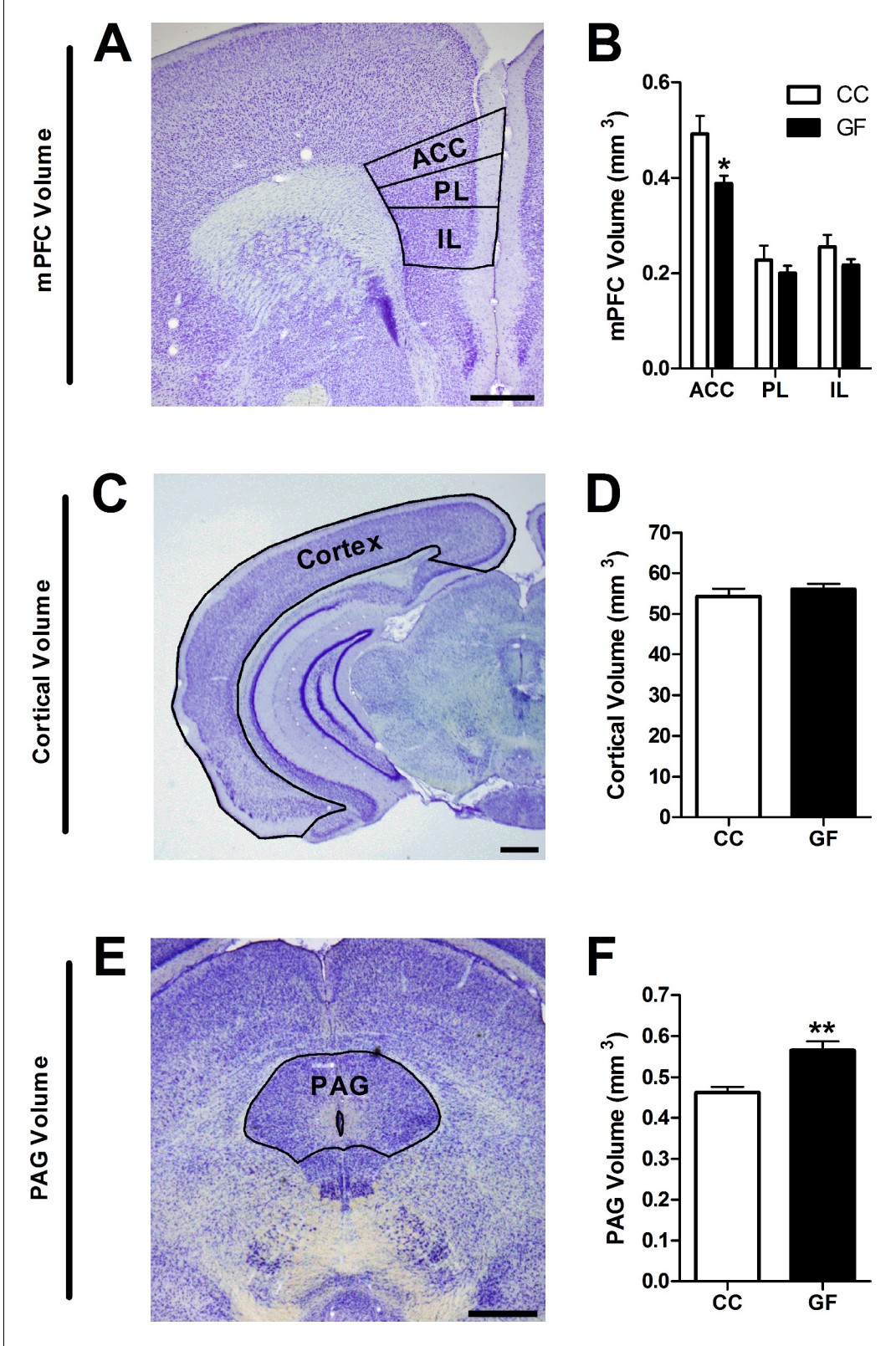

**Figure 3.** Reduction in ACC and increase in PAG volume in GF mice. (A,C,E), Representative thionin-stained section of the mPFC (A), cortex (C), and PAG (E). The volumes of the defined (black lines) subregions of interest were estimated using Cavalieri's principle. Scale bars = 0.5 mm. (B) In GF mice, the volume of the ACC was reduced. (D) Cortical volume did not differ between CC and GF mice. (F) The PAG was larger in GF versus CC mice. CC, n = 5; GF, n = 6–7.

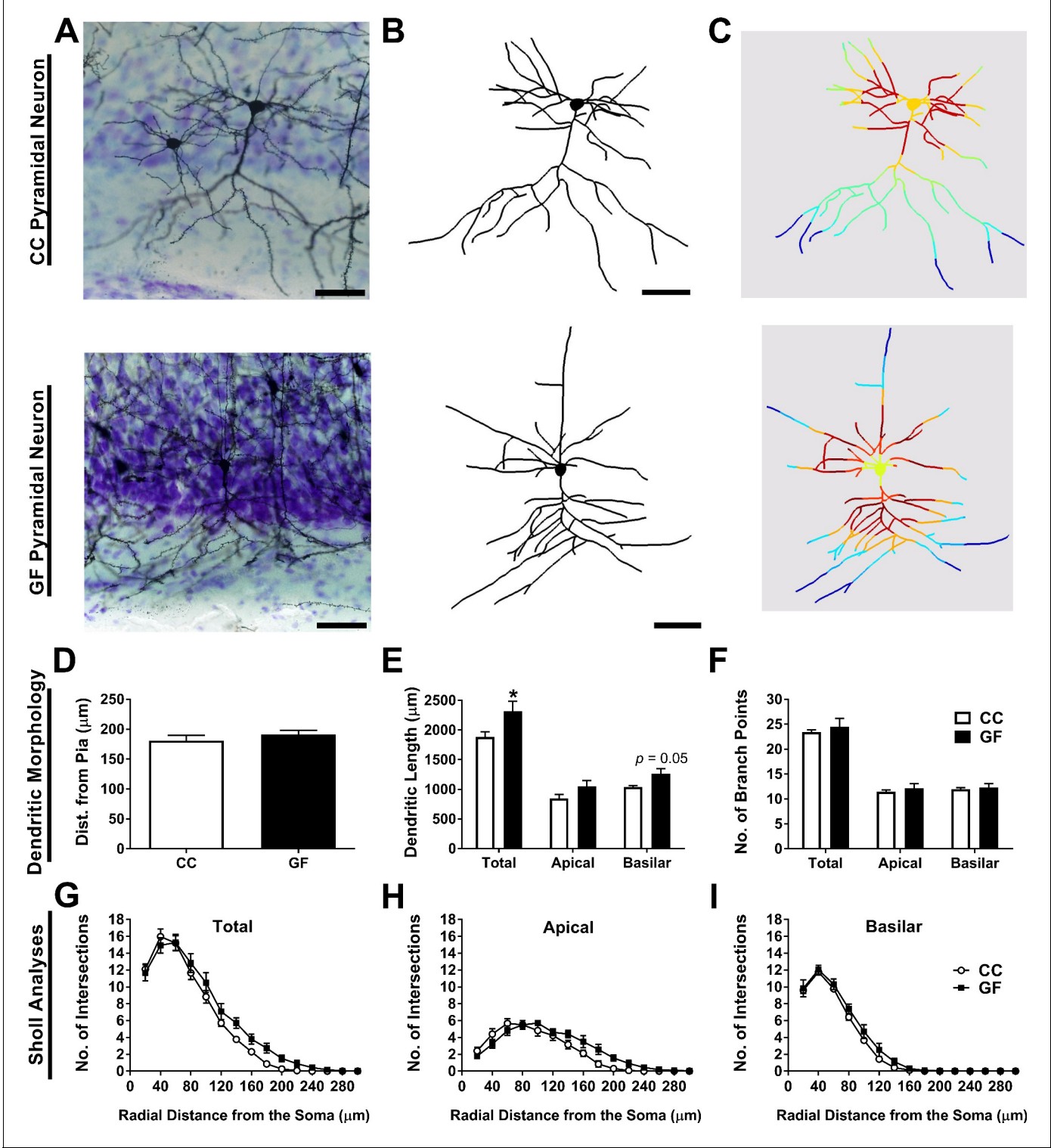

**Figure 4.** Basilar dendritic elongation in ACC pyramidal neurons of GF mice. (A,B,C) Representative images of Golgi-stained layer II/III pyramidal neurons of CC and GF mice (A). Neurons were reconstructed in 3D using morphometric software (B). Sholl analysis was performed on 2D renderings of the neurons (incremental radii from the soma are indicated by the color gradient; C). Scale bars = 50 μm. (D) There was no group difference in the topographical location (i.e. cell layer) of the neurons in the ACC. (E) When compared to controls, the dendrites of ACC pyramidal neurons of GF mice were longer. This elongation was principally localized to the basilar dendritic arbor. (F) There was no change in the number of branch points of ACC pyramidal neurons between groups. (G) Sholl analysis of the total dendritic arbor revealed no difference in dendritic complexity between CC and GF

*Figure 4 continued on next page*

*Figure 4 continued*
mice. (H) In GF mice, the apical dendritic arbor showed altered dendritic complexity; however, *post hoc* comparisons revealed no statistically significant distances in which this change occurred. (I) There was no group difference in the complexity of basilar dendrites. For both CC and GF mice, n = 5.
The following figure supplement is available for figure 4:

**Figure supplement 1.** Increase of thin and stubby spines on ACC pyramidal neurons of GF mice.

## Altered volume of pain-related brain structures of GF mice

We compared the volumes of the mPFC subregions in CC versus GF mice (*Figure 3A and B*). The anterior cingulate cortex (ACC) was 21% smaller in GF mice compared to controls ($t_{10}$ = 2.78, p=0.020). There were no significant differences in the volumes of the prelimbic (PL) and infralimbic (IL) cortices between groups (PL: $t_{10}$ = 0.90, p=0.39; IL: $t_{10}$ = 1.56, p=0.15).

To determine if the reduction in ACC volume observed in GF mice extended to the rest of the cortex, we measured cortical volume (*Figure 3C and D*). Total cortical volume was not significantly different in CC and GF mice ($t_9$ = 0.77, p=0.46).

We measured periaqueductal grey (PAG) volume to determine if brain regions in the descending pain modulation pathway were affected by microbial status (*Figure 3E and F*). The PAG was 22% larger in GF compared to CC mice ($t_9$ = 3.83, p=0.0040).

## Dendritic hypertrophy of ACC pyramidal neurons of GF mice

### Dendritic morphology

Morphometric analyses were performed on the dendrites of Golgi-stained layer II/III ACC pyramidal neurons (*Figure 4A–I*). There was no significant difference in the distance from the pia for neurons from CC and GF mice ($t_8$ = 0.96, p=0.36; *Figure 4D*), indicating that neurons from both groups were located in similar topographical locations (i.e. cell layer) in the ACC. There was no statistically significant difference in total dendritic length ($t_8$ = 2.28, p=0.052; *Figure 4E*) or the number of branch points ($t_8$ = 0.59, p=0.57; *Figure 4F*) between groups. The basilar dendrites of GF mice were 22% longer than controls, but there was no significant change in apical dendritic length (apical: $t_8$ = 1.72, p=0.12; basilar: $t_8$ = 2.65, p=0.029; *Figure 4E*). There was no significant group difference in the number of branch points in either apical or basilar dendrites (apical: $t_8$ = 0.73, p=0.49; basilar: $t_8$ = 0.39, p=0.71; *Figure 4F*).

2D Sholl analysis revealed no significant group differences in ACC pyramidal neuron total dendritic distribution (main effect $F_{1, 8}$ = 2.16, p=0.18; interaction $F_{14, 112}$ = 1.43, p=0.15; *Figure 4G*). We then conducted more specific Sholl analyses, investigating the complexity of the apical and basilar dendritic arbors separately. Sholl analysis revealed a significant interaction of group (CC vs GF) and distance from the soma on the distribution of apical dendrites ($F_{14, 112}$ = 3.06, p<0.001); however, post hoc comparisons uncovered no statistically significant dendritic regions in which the change in complexity occurred (*Figure 4H*). There was no significant difference in the complexity of the basilar dendrites (main effect $F_{1, 8}$ = 3.29, p=0.11; interaction $F_{14, 112}$ = 0.67, p=0.80; *Figure 4I*).

### Spine density and subtype analysis

The spine subtypes and density of each layer II/III ACC pyramidal neuron was determined by sampling both apical and basilar dendritic segments (*Figure 4—figure supplement 1A–I*). There was no significant difference in spine width on any portion of the dendritic arbor (overall: $t_8$ = 0.16, p=0.88; apical: $t_8$ = 0.16, p=0.88; basilar: $t_8$ = 0.16, p=0.88; *Figure 4—figure supplement 1D*). Similarly, there was no significant group difference in spine length overall, on apical or basilar dendrites (overall: $t_8$ = 0.23, p=0.82; apical: $t_8$ = 0.38, p=0.72; basilar: $t_8$ = 0.83, p=0.43; *Figure 4—figure supplement 1E*). The overall spine density (includes all spine subtypes) was 26% higher in GF mice ($t_8$ = 3.46, p=0.0085; *Figure 4—figure supplement 1F*). This increase in spines occurred on all portions of the dendritic arbor: spine density was increased by 23% and 31% on apical and basilar dendrites, respectively (apical: $t_8$ = 3.22, p=0.012; basilar: $t_8$ = 3.11, p=0.014; *Figure 4—figure supplement 1F*). GF mice had 20% more thin spines overall on ACC pyramidal neurons compared to controls ($t_8$ = 2.50, p=0.037; *Figure 4—figure supplement 1G*). This increase in thin spines

appears to be principally located on basilar dendrites: there was a strong trend toward an increase in thin spines on basilar dendrites but no such effect on apical dendrites (apical: $t_8$ = 1.73, p=0.13; basilar: $t_8$ = 2.20, p=0.060; *Figure 4—figure supplement 1G*). In GF mice, the overall stubby spine density was increased by 53% compared to controls ($t_8$ = 3.30, p=0.011; *Figure 4—figure supplement 1H*). The higher stubby spine density was located to the basilar dendritic arbor: the basilar dendrites of GF mice had 69% more stubby spines while there was no change in apical stubby spine density (apical: $t_8$ = 2.04, p=0.076; basilar: $t_8$ = 4.77, p=0.0014; *Figure 4—figure supplement 1H*). There was no significant group difference in mushroom spine density on any portion of the dendritic arbor (overall: $t_8$ = 1.55, p=0.16; apical: $t_8$ = 1.53, p=0.16; basilar: $t_8$ = 1.53, p=0.16; *Figure 4—figure supplement 1I*). Finally, there was no group difference in filopodia spine density overall, or on apical and basilar dendrites (overall: CC = 0.041 ± 0.0067, GF = 0.065 ± 0.015, $t_8$ = 1.43, p=0.19; apical: CC = 0.037 ± 0.010, GF = 0.065 ± 0.016, $t_8$ = 1.43, p=0.19; basilar: CC = 0.044 ± 0.0051, GF = 0.065 ± 0.014, $t_8$ = 1.33, p=0.22).

## Does microbial colonization normalize visceral hypersensitivity, spinal cord TLR and cytokine expression in GF mice?

### Reversal of visceral hypersensitivity in GF mice following microbial colonization

To determine whether microbial colonization later in life could reverse the visceral hypersensitivity observed in GF mice, a cohort of GF mice were exposed to feces from CC mice. Microbial colonization normalized visceral pain perception in response to colonic stimulation (main effect of pressure: $F_{4,68}$ = 33.42, p<0.001; main effect of group: $F_{2,17}$ = 11.85, p<0.001; *Figure 5A*). The pain threshold was also normalized in GFC mice ($F_{2,25}$ = 4.56, p=0.021; *Figure 5B*).

### Normalization of gene expression in the spinal cord of GF mice following microbial colonization

GFC animals displayed normalized gene expression levels of TLR1 ($F_{2,46}$ = 7.35, p=0.0018), TLR2 ($F_{2,45}$ = 5.77, p=0.0060), TLR4 ($F_{2,47}$ = 6.05, p=0.0047), TLR5 ($F_{2,45}$ = 3.98, p=0.026), TLR7 ($F_{2,44}$ = 6.96, p=0.0024), and TLR12 ($F_{2,45}$ = 4.64, p=0.015). Although microbial colonization normalized the expression levels of TLR3 ($F_{2,45}$ = 4.04, p=0.025) and TLR9 ($F_{2,46}$ = 3.54, p=0.038) in relation to CC mice, no significant differences were observed between GFC and GF mice (*Figure 6A–H*).

Similarly, microbial colonization normalized the gene expression levels of IL6 ($F_{2,27}$ = 6.12, p=0.0068), TNFα ($F_{2,42}$ = 4.38, p=0.019), and IL10 ($F_{2,43}$ = 6.01, p=0.0050). Microbial colonization normalized the expression of IL1α ($F_{2,46}$ = 3.52, p=0.038) and IL1β ($F_{2,37}$ = 7.10, p=0.0026)

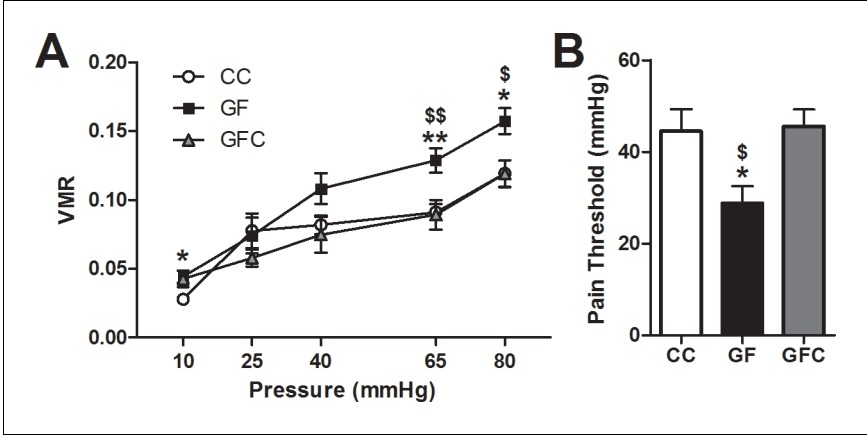

**Figure 5.** Normalization of visceral hypersensitivity following colonization of GF mice. (A,B) Microbial colonization restored normal visceral pain responsivity (A) and pain threshold (B). For this and subsequent figures, *p<0.05, **p<0.01, and ***p<0.001 versus CC mice; $p < 0.05; $$p < 0.01 versus GFC mice. CC, n = 10; GF, n = 8; GFC, n = 9.

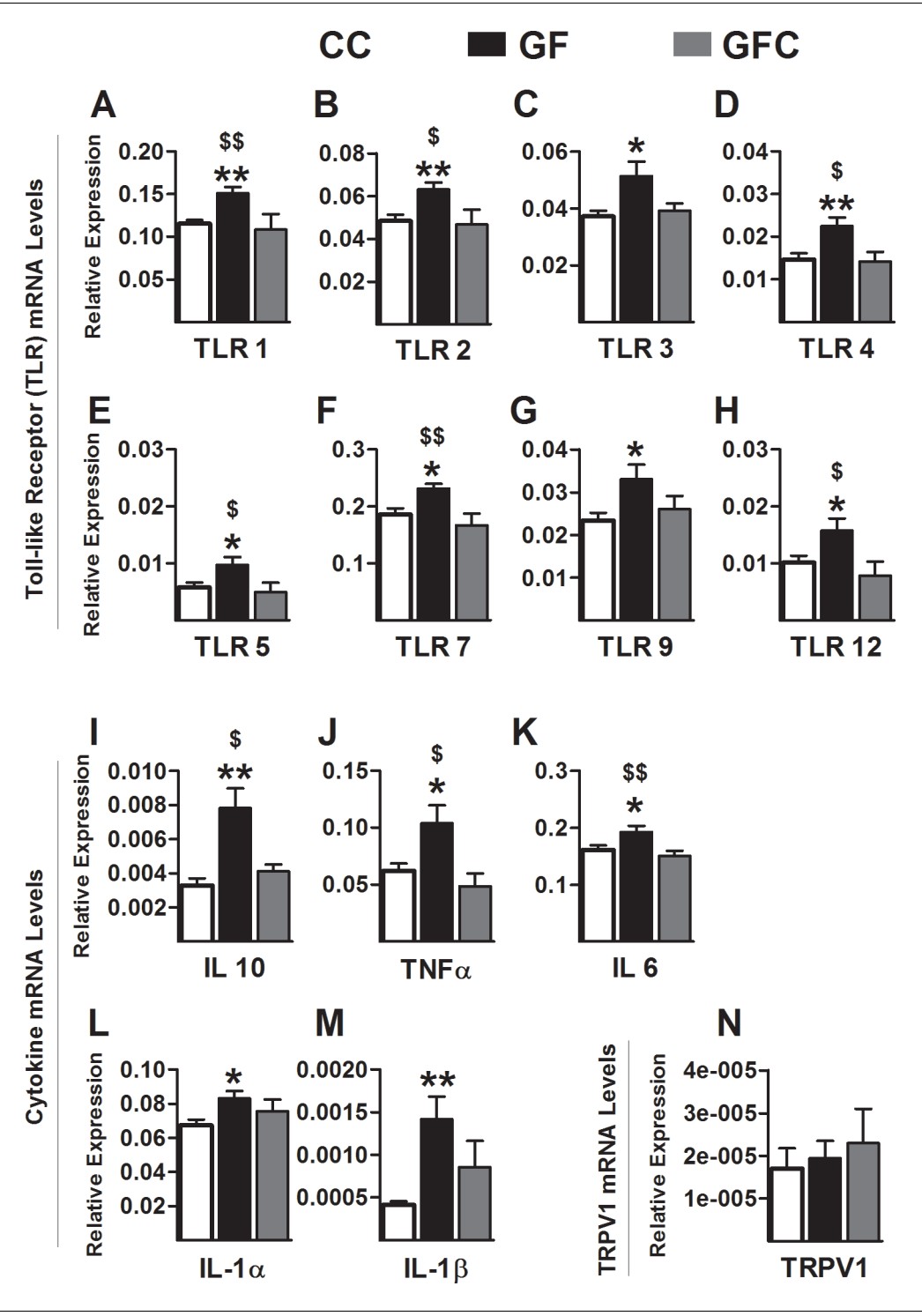

**Figure 6.** Normalization of toll-like receptor and cytokine gene expression in the spinal cord of GF mice following microbial colonization. (A–H) Microbial colonization normalized the increased gene expression levels of the toll-like receptors TLR1 (A), TLR2 (B), TLR4 (D), TLR5 (E), TLR7 (F), and TLR12 (H) observed in the spinal cord of GF mice. No changes were observed in the expression levels of TLR3 (C) and TLR9 (G) between GFC and GF mice. (I–M) Microbial colonization reversed the elevated gene expression of the cytokines IL6 (I), TNFα (J), and IL10 (K) in the spinal cord of GF mice. No changes were observed in the expression levels of IL1α (L) and IL1β (M) between GFC and GF mice. (N) No group changes were observed in the expression levels of TRPV1. CC, n = 16–20; GF, n = 15–20; GFC, n = 7–10.

compared to controls; however, there was no statistical difference between GFC and GF mice (*Figure 6I–M*). In addition, gene expression levels of TRPV1 were measured. No significant changes were observed between groups ($F_{2, 20}$ = 0.28, p=0.7621; *Figure 6N*).

## Reversal of astrocyte and microglial activation in the spinal cord of GF mice following microbial colonization

Because TLRs are predominantly expressed on astrocytes and microglia within the CNS (*Lehnardt et al., 2003*), we investigated the relationship between glia and the enhanced gene expression of several TLRs in the spinal cord. Similarly to TLRs, protein levels of GFAP and Cd11b, respective markers of astrocyte and glial activation (*Yao et al., 2014*; *Bignami et al., 1972*), were increased in the spinal cord of GF versus CC mice. Importantly, microbial colonization normalized

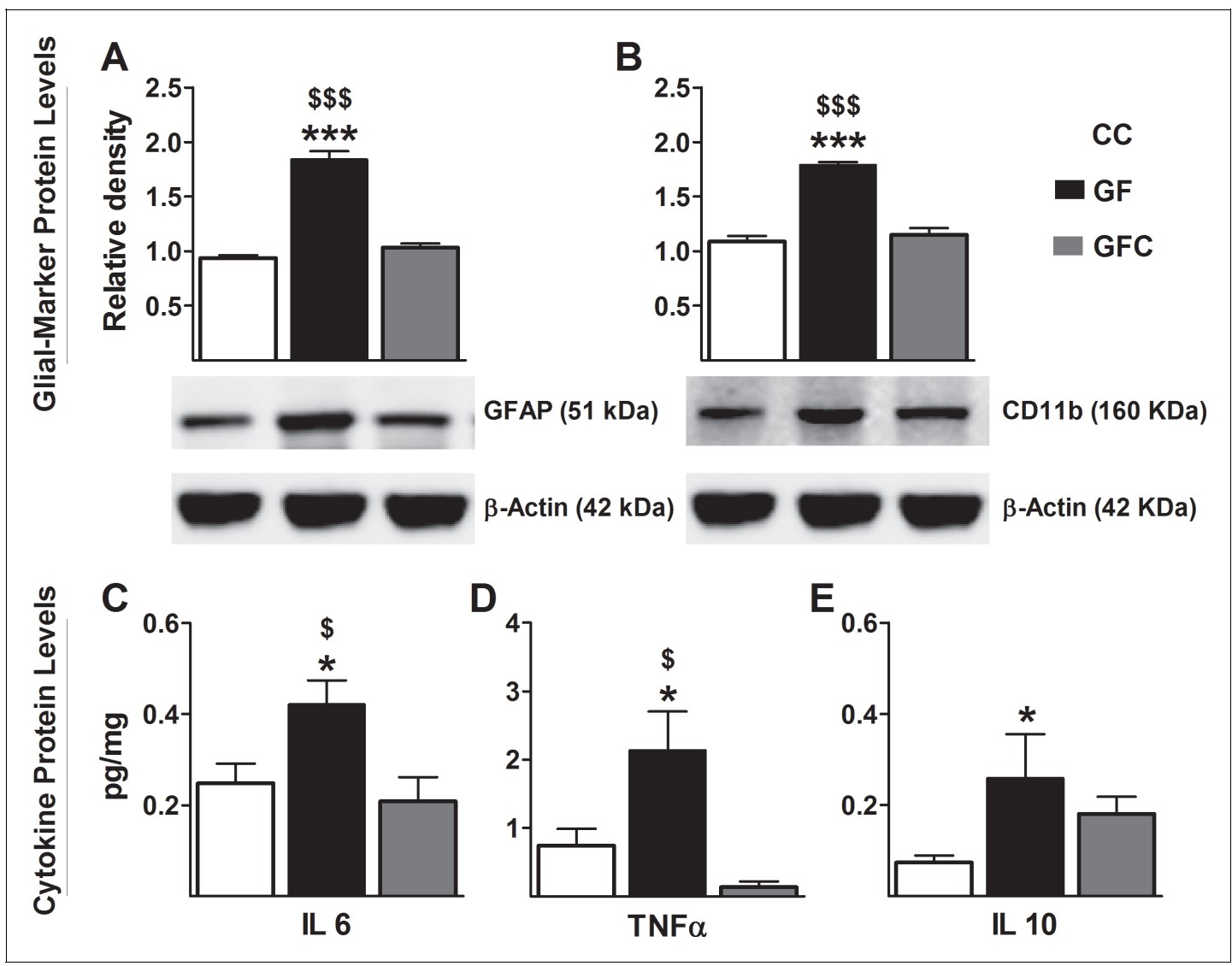

**Figure 7.** Normalization of glial activation and cytokine protein expression in spinal cord of GF mice following microbial colonization. Western blot analysis was performed for GFAP (astrocyte marker) and Cd11b (microglial marker) in the lumbosacral region of the spinal cord in CC, GF, and GFC mice. (A,B), The increased expression of both GFAP (**A**) and CD11b (**B**) in GF mice was normalized following colonization. (C–E) ELISA assays were performed to assess the protein levels of cytokines. Microbial colonization similarly normalized the elevated protein levels of the cytokines IL6 (**C**) and TNFα (**D**) in the spinal cord. However, no change was observed in the expression of IL10 (**E**) between GFC and GF mice. CC, n = 5–11; GF, n = 6–10; GFC, n = 4–8.

the protein levels of GFAP ($F_{2,17}$ = 79.6, p<0.001; *Figure 7A*) and CD11b ($F_{2,19}$ = 47.47, p<0.001; *Figure 7B*) to control levels.

## Normalization of protein levels of cytokines in the spinal cord of GF mice following microbial colonization

We measured protein levels of cytokines to confirm that the microbiota was regulating their expression in the spinal cord. Expression levels of IL6 ($F_{2,23}$ = 4.70, p=0.021, *Figure 7C*) and TNFα ($F_{2,24}$ = 4.79, p=0.019; *Figure 7D*) were significantly increased in GF mice compared to CC mice and normalized following the microbial colonization. Although microbial colonization returned the protein expression of IL10 to baseline in relation to CC mice ($F_{2,23}$ = 3.74, p=0.041, *Figure 7E*), no significant differences were observed between GFC and GF mice. The values for IL1α and IL1$\beta$ were below detection range. No significant differences were observed in the levels of expression of IL12 (CC: 0.6341 ± 0.158, GF: 0.8422 ± 0.142, GFC: 0.7896 ± 0.121, $F_{2, 21}$ = 0.58, p=0.57) and IFNγ (CC: 0.0141 ± 0.0038, GF: 0.1271 ± 0.0013, GFC: 0.0137 ± 0.0041, $F_{2, 24}$ = 0.19, p=0.82) between groups.

## Discussion

The aim of this study was to determine if commensal gut bacteria are required for normal visceral sensitivity and CNS pain processing. Our results show that mice raised in GF conditions exhibited visceral hypersensitivity and an increase in cytokine and TLR expression in the spinal cord. We have also demonstrated that postnatal microbial colonization normalizes the visceral hypersensitivity and spinal cord inflammatory profile of GF mice. In addition, we found that these animals had increased ACC and decreased PAG volumes as well as dendritic hypertrophy of single pyramidal neurons in the ACC. Taken together, these results clearly indicate that the gut microbiota is required for normal visceral nociception and pain processing at the level of the CNS.

### The microbiota and visceral pain

This study adds proof-of-principle evidence to the growing body of support for an association between the gut microbiota and visceral nociception (*Eisenstein, 2016*). We have previously shown that exposure to stressors or antibiotic treatment in early life can have enduring effects on visceral pain in rodents, and these effects are accompanied by a change in microbial composition (*O'Mahony et al., 2014*, *2009*). Moreover, in rodents, probiotics can ameliorate visceral pain induced by stress (*Gareau et al., 2007*; *McKernan et al., 2010*; *Ait-Belgnaoui et al., 2006*) or antibiotic administration (*Verdú et al., 2006*), and even exert an inhibitory influence on visceral sensitivity in healthy rats (*Kamiya et al., 2006*). Clinical studies have repeatedly found that the composition of the gut microbiota is altered in patients with IBS (*Collins, 2014*; *Jeffery et al., 2012*) and that patients suffering from abdominal pain can benefit from probiotic treatment (*Clarke et al., 2012*). Although these studies report an association between altered microbial composition and visceral pain processing, the impact of the microbiota cannot be separated from other factors, such as stress or antibiotic side effects. In this set of experiments, we use GF as a tool to show that visceral sensitivity is heightened in mice in the absence of any microbiota.

Although it is now clear that the gut microbiota plays a role in visceral nociception, the exact mechanism by which the microbiota exerts its influence on the CNS remains unknown. It is likely that many systems simultaneously contribute to the altered nociception in GF mice, including immune, neural, endocrine, and metabolic processes (*Sampson and Mazmanian, 2015*; *Dinan and Cryan, 2012*; *Mayer, 2011*; *Luczynski et al., 2016*; *Cryan and Dinan, 2012*). Indeed, GF mice have blunted immune function and altered TLR expression in the gut (*Shanahan, 2002*; *Clarke et al., 2013*). Interestingly, the development in early life and function of the enteric nervous system (ENS) in adulthood is altered in GF mice (*Collins et al., 2014*; *McVey Neufeld et al., 2013*). Enteric sensory neurons synapse with vagal nerve endings in the gut (*Perez-Burgos et al., 2014*), providing a direct route whereby the intestinal bacterial status can be communicated to the brain. Growing up germ-free heightens hypothalamic-pituitary-adrenal axis responsivity, which is thought to occur via a neuronal mechanism (*Sudo et al., 2004*). GF rodents also have a reduced production of short chain fatty acids (*den Besten et al., 2013*), which are metabolites produced by gut bacteria thought to be key

mediators of gut-brain signaling. Moving forward, it would be interesting to determine if is it the cell surface proteins on intact bacteria or their metabolites and other products that are involved in changes along the brain-gut axis.

## Spinal cord signaling of visceral pain

Within the gastrointestinal tract, nociceptors respond to many stimuli including stretch, pH, bacterial products, immune signaling molecules, and neurotransmitters released from the ENS or from the bacteria themselves (*Sengupta, 2009*). From the gut, nociceptive signals are transmitted to the spinal cord, and subsequently the brain. Recent evidence has implicated neuroimmune and spinal microglial mechanisms in chronic visceral pain (*Ji et al., 2013*; *Grace et al., 2014*; *Bradesi, 2010*). TLR receptor signaling is also involved in visceral nociception and IBS pathology (*Tramullas et al., 2014*, *Tramullas et al., 2016*). Moreover, microglial structure and homeostasis are disrupted in both GF and antibiotic-treated mice (*Erny et al., 2015*), indicating that the microbiota is required for normal microglial function in the CNS. Microglia can be activated through TLR signaling, and once activated, increase the secretion of various cytokines (*Bradesi, 2010*). In the present study, we report that glial activation, TLR expression, and cytokine signaling are all increased in the lumbosacral spinal cord of GF mice, an area associated with signals from the colon. Indeed, microglial activation (*Saab et al., 2006*), reduced glutamate-reuptake by astrocytes (*Gosselin et al., 2010*), and increased TLR signaling (*Tramullas et al., 2014*) in the spinal cord have been previously shown in rodent models with visceral hypersensitivity.

TRPV1 receptors are broadly expressed in the gastrointestinal tract and many areas of the CNS, including the spinal cord and is well-recognized as a transducer of noxious stimuli (*Nagy et al., 2004*). We found no changes in the mRNA expression of the TRPV1 receptor in the GF spinal cord. This result is in line with previous data from our group showing that the colon of GF mice exhibits the same responsiveness to capsaicin (a TRPV1 agonist) relative to controls (*Lomasney et al., 2014*), suggesting no change in the expression of the TRPV1 receptor. In contrast, rats exposed to vancomycin in early in life resulting in an altered gut microbiome, show enhanced visceral pain perception and a decrease in spinal cord TRPV1 expression (*O'Mahony et al., 2014*). In the gut, TRPV1 receptors play a key role in pain perception (*Holzer, 2011*); however, TRPV1 signaling does not appear to be responsible for the visceral hypersensitivity observed in GF mice. Further studies are required to confirm changes in the expression of TRPV1 receptor in other areas of the CNS involved in visceral pain perception. With regard to what influences the changes in cytokines and TLRs in the spinal cord of GF mice we can speculate that lack of certain metabolites and or short chain fatty acids which reduce visceral pain or exaggerated sensation in a conventional mouse are not present to gate the pain signals which can lead to altered signaling in the spinal cord. Moreover, GF are known to have an altered immune system and hence changes in spinal cord immune players fits well with the literature.

## Prefrontal cortical structural changes and visceral pain

The affective or emotional component of pain is mediated by the ACC (*Apkarian et al., 2005*). Studies in humans with IBS (*Tillisch et al., 2011*; *Mertz et al., 2000*) and in animal models (*Gibney et al., 2010*; *Felice et al., 2014*; *Bliss et al., 2016*) have revealed increased activation in the mPFC in response to visceral pain. In addition, imaging studies have consistently observed reduced cortical grey matter in patients with IBS (*Davis et al., 2008*). Although cortical structure has yet to be investigated in animal models of visceral pain, rodents with long-lasting neuropathic pain show a reduction of ACC volume (*Seminowicz et al., 2009*) and basilar dendritic hypertrophy of pyramidal neurons (*Metz et al., 2009*). We report a remarkably similar reduction in ACC volume in GF mice as well as elongation of single ACC pyramidal neurons. Importantly, GF mice showed no difference in mushroom spines, which represent mature, long-lasting postsynaptic connections (*Matsuzaki et al., 2004*). Instead, the density of 'immature' thin and stubby spines was higher in these animals. These results can be interpreted as a microbiota-induced deficit in synaptic pruning resulting in the hyperactivity of ACC neurons. Such changes in ACC signaling would likely impact visceral sensitivity, as this area receives and sends projections to many pain-relevant brain areas including the limbic system, ventral tegmental area and the PAG (*Hoover and Vertes, 2007*).

The neurons in the PAG, once activated are involved in an endogenous pain inhibitory system (*Apkarian et al., 2011*). It is currently unclear what the relative contribution of the altered PAG volume is to the visceral hypersensitivity behavior observed in GF mice. The PAG receives signals from cortical areas such as the ACC where this connection is thought to mediate expectation and placebo analgesia (*Petrovic et al., 2002*). Hence, input changes from the ACC may have a reciprocal effect on the PAG. What is evident, however, is that both cortical and subcortical pain pathways are altered in GF mice. Taken together, our results suggest that gross morphological and ultrastructural changes in the ACC and PAG could underlie the visceral hypersensitivity observed in GF mice.

## Conclusions

Our results show that the gut microbiota dramatically impacts visceral sensitivity and affects signaling in key structures involved in the processing and integration of painful stimuli. These results further reinforce the idea that the microbial composition of mice in conventional facilities may affect the responsiveness to CRD (*Verdú et al., 2006*). Importantly, this maladaptive pain responsivity is amenable to microbial-based interventions in later life. This is noteworthy as numerous postnatal colonization experiments have been performed in GF mice, with certain behavioral and physiological changes proving reversible (*Clarke et al., 2013*; *Sudo et al., 2004*; *Diaz Heijtz et al., 2011*) and others irreversible (*Desbonnet et al., 2014*; *Stilling et al., 2015*). Hence the addition of colonization experiments with regard to morphological measures would be of interest in future studies. Moreover, we find across a number of different domains that the effects of the microbiome on brain and behavioural are much more robust in males than females (*Clarke et al., 2013*; *Desbonnet et al., 2015*; *Hoban et al., 2016*). Therefore, future analysis of visceral sensitivity in female animals is indeed worthy of investigation. Improving our understanding of the impact of microbiota on visceral pain sensitivity (*O'Mahony et al., 2017*) may ultimately inform novel therapies for the treatment of gastrointestinal pain disorders.

## Materials and methods

### Animals

Swiss Webster breeding pairs for both GF and CC mice were supplied by Taconic (Germantown, New York, USA) and first-generation male offspring were studied in all experiments. In the University College Cork GF Unit, GF mice were group housed in flexible film gnotobiotic isolators maintained at $21 \pm 1°C$, with 55–60% relative humidity, under a 12 hr light/dark cycle. A group of GF mice were colonized (GFC) on postnatal day 21 with microbiota obtained from CC mice. Briefly, GF mice were removed from the GF facility and allowed to grow to adulthood in the conventional animal facility in cages with bedding and fecal matter from CC mice, a method which has previously been shown to effectively restore a normal microbiota (*Clarke et al., 2013*; *Desbonnet et al., 2014*; *O'Connell Motherway et al., 2011*). CC and GFC mice were group housed in the standard animal facility which was held at the same controlled conditions and light/dark cycle as the GF mice. GF, CC, and GFC mice were age-matc and fed the same autoclaved pelleted diet (Special Diets Services, product code 801010) and were housed 2–5 per cage (only one group of CC mice was housed two per cage). Animals were tested or euthanized at 5–10 weeks of age. All experiments were performed in accordance with the guidelines of European Directive 86/609/EEC and the Recommendation 2007/526/65/EC and were approved by the Animal Experimentation Ethics Committee of University College Cork.

Four cohorts of animals were used to perform the experiments in this paper. Animals in the first cohort (CC = 10, GF = 9; *Figure 1*) underwent CRD and then were euthanized. No samples were collected from this cohort. Animals in the second cohort (CC and GF = 9–10; *Figure 2*) were sacrificed without anesthesia and spinal cord tissue was collected to perform gene expression analyses. These mice were not exposed to CRD. After data analysis, we decided that it would be of interest to determine if bacterial colonization could rescue the visceral hypersensitivity and associated changes noted in GF mice from the first and second cohorts. With the third cohort of mice (CC = 20, GF = 20, GFC = 10; *Figures 5*, *6* and *7*)), we repeated the CRD and collected the spinal cord tissue. Due to the limited number of GFC mice available, most GFC animals used in the spinal cord analysis had been exposed to CRD. However, we verified that there was no statistical difference between those

animals that underwent CRD and those that did not in the CC and GF groups. Animals in the fourth cohort (CC = 17, GF = 18; *Figures 3* and *4*, and 4 supplement 1) were euthanized without undergoing any procedures. Brain tissue from 10 animals from each group was processed for Golgi-Cox staining and the remainder of the brains were processed as per the stereology protocol.

## Colorectal distention (CRD)

CRD was carried out as previously described (*O'Mahony et al., 2012*; *Tramullas et al., 2012*). The CRD-system was composed of a barostat (Distender Series II, G and J Electronics, Toronto, ON, Canada) and a transducer amplifier (LabTrax 4, World Precision Instruments, Sarasota, FL). A custom-made balloon (2 cm length x 1 cm inflated diameter) prepared from a polyurethane plastic bag (GMC Medical, Denmark) was tied over a PE60 catheter with silk 4.0. Before securing the balloon to the catheter, several holes were punched in the distal 20 mm of the tubing with a 27-gauge needle to allow the balloon to inflate.

On the experimental day, mice were lightly anesthetized with isoflurane (2% vapor in oxygen; Iso-Flo, Abbott, UK) and a lubricated balloon with a connecting catheter was inserted into the colon, 0.5 cm proximal to the anus. The catheter was fixed to the base of the tail with tape to avoid any displacement. Unrestrained mice were allowed to recover for 10 min before starting the CRD procedure. The balloon was connected to the barostat system and subsequent pressure changes within the distending balloon, observed in response to a distension paradigm, were monitored and recorded using Data Trax two software (World Precision Instruments, Sarasota, FL). The ascending phasic distension (from 10 to 80 mmHg) paradigm, consisting of three 20 s pulses at each pressure and 5-min inter-pulse intervals, was used. The visceromotor responses (VMR) were quantified as pressure changes observed within the colonic distending balloon during the colorectal distension procedure. VMR were calculated as the average of the three consecutive pulses for each pressure. Additionally, for each animal, pain threshold was defined as the pressure which exceeded the mean baseline activity plus three times the standard deviation (see *Figure 1*-supplement 1A 4 for phasic stages). At the end of the experiments, the balloon was carefully removed and the animals were returned to their home cages.

## Spinal cord analysis

### Sample collection

Mice were sacrificed without anesthesia, the spinal cord was extracted by hydro-extrusion, and the dorsal root ganglia were removed (*Lantero et al., 2014*). The lumbosacral region of the spinal cord was dissected, snap-frozen and stored at −80°C. The lumbosacral segments of the spinal cord receive input from visceral afferents originating from the colon (*Kyloh et al., 2011*). We did not collect dorsal root ganglia or gut mucosa.

### Qualitative real-time PCR

Total RNA was extracted from the spinal cord using the Qiagen RNeasy Lipid Mini Kit (QIAGEN, Valencia, CA). RNA quality was assessed using the Agilent Bioanalyzer (Agilent, Stockport, UK) according to the manufacturer's procedure and an RNA integrity number (RIN) was calculated. RNA with RIN value >7 was used for subsequent experiments. Complementary DNA was synthesized using 1 mg total RNA using random primers. Quantitative PCR was carried out in an LightCycler480 System using specific probes (six carboxy fluorescein - FAM) designed by Applied Biosystems to mouse specific TLR1, TLR2, TLR3, TLR4, TLR5, TLR6, TLR7, TLR8, TLR9, TLR11, TLR12, TNF$\alpha$, IL1$\alpha$, IL1$\beta$, IL6, IL10, IL12a, IL12b, IFN$\gamma$, and TRPV1, while using $\beta$-Actin as an endogenous control. Data were analyzed using the $2^{-\Delta Ct}$ method and expressed as relative expression. No significant differences were observed in the mRNA expression levels of $\beta$-Actin between groups.

### Western blot

Western blot was performed as previously described (*Tramullas et al., 2010*). Whole-cell lysates were prepared from the lumbosacral region of the spinal cord and total protein was determined using the Quan-iT protein assay kit (Invitrogen, Ireland). Equal amounts of protein were subjected to electrophoresis on 4–12% gradient gels (NuPAGE, Invitrogen, Ireland) and transferred to a polyvinylidene difluoride membrane (Bio Rad, Ireland). Membranes were then incubated with goat anti-

GFAP (1:1000; Sigma, Ireland), the astrocyte marker, rabbit anti-CD11b (1:500; Novus Biologicals, UK), a microglia activation marker; and mouse anti $\beta$-actin (1:15000; Sigma, Ireland). Immunoreactivity was detected with Pierce ECL detection reagent (Thermo Scientific, IL, USA) and visualized using a luminescent image analyzer (LAS-3000, Fujifilm, Ireland). The relative density of the immunoreactive bands was quantified using Fiji Is Just ImageJ (FIJI) software (*Schindelin et al., 2012*) and normalized to the relative density of mouse anti-$\beta$ actin (Sigma, Ireland) for each sample. No significant differences were observed in the protein expression levels of $\beta$-Actin between groups.

## Cytokine assay

Measurement of cytokine levels on the spinal cord was carried out with custom mouse Multi-spot 96-well plates (Meso Scale Discovery, Rockville, MD) according to instructions of the manufacturer. Briefly, samples were weighted, homogenized with buffer and added to pre-coated wells in duplicate and incubated for 2 hr. Wells were then washed using phosphate buffered saline-Tween 20 and incubated with an antibody mix ($\alpha$-IL6, $\alpha$-TNF$\alpha$ and $\alpha$-IL10 antibodies) in diluent for 2 hr. ELISA plates were analyzed using the Sector 2400 imager from Meso Scale Discovery. This is an ultra-sensitive method which has a detection limit of 0.3 pg/ml. Data were normalized to the amount of tissue per well and expressed as pg/mg.

## Stereological measurement of brain volume

The volumes of the mPFC, total cortex, and periaqueductal grey (PAG) were estimated using Cavalieri's principle (*Gundersen et al., 1988*). Brain structures were characterized using the *Paxinos and Franklin, 2001* atlas as a guide.

## Histological preparation

Animals were terminally anesthetized with sodium pentobarbital and perfused transcardially with 0.1 M phosphate buffer, followed by 4% paraformaldehyde. Brains were postfixed in a 4% paraformaldehyde solution for 24 hr and then cryoprotected in a 30% sucrose solution for a further 48 hr. The brains were flash-frozen in isopentane and stored at −80°C. Using a cryostat, brains were coronally sectioned at 40 µm and sections were lightly stained with thionin. All slides were coded to obscure the experimental group of each animal until statistical analysis.

## Analysis of medial prefrontal cortical volume

The volumes of the anterior cingulate (ACC; Cg1 + Cg2), prelimbic (PL), and infralimbic (IL) cortices were estimated. The first rostrocaudal appearance of the corpus callosum (Bregma +1.98 mm) was chosen as the sample starting point and its decussation was chosen as the end point (Bregma +1.10 mm). The left and right cortices were imaged separately to allow for better resolution and visualization of cell layers and total cortical and mPFC volume were calculated by summing the hemisphere values. For each animal, 12 to 16 evenly-spaced serial sections with a randomized start were studied (periodicity of every slice; section increment of 0.04 mm). The region of interest was digitally outlined in each section (4× magnification; N.A. 0.1) and converted to area values (mm$^2$) using FIJI software (*Schindelin et al., 2012*). The area values were then combined with the cut section thickness (0.04 mm) and section increment to compute the volumes of the prefrontal cortex subregions.

## Analysis of total cortical volume

The volume of the total cortex was estimated using the boundaries previously defined by *Smiley et al. (2015)*. The first rostrocaudal appearance of the corpus callosum (Bregma +1.98 mm) was chosen as the sample starting point and the termination of the hippocampus was chosen as the end point (disappearance of the granular layer of the dentate gyrus; Bregma −4.03 mm). The analysis excluded the mPFC and the orbitofrontal cortex. The left and right cortices were imaged separately to allow for better resolution and visualization of cell layers and total cortical and mPFC volume were calculated by summing the hemisphere values. For each brain, 24 to 30 evenly-spaced serial sections with a randomized start were studied (periodicity of every fourth slice; section increment of 0.16 mm). The perimeter of the cortex was digitally outlined in each section (2× magnification, N.A. 0.08) and volume was calculated. One GF mouse was excluded from cortical volumetric analysis due to damaged tissue.

There were no significant hemispheric differences in mPFC or total cortical volume between mice of the same group (p>0.05 for all subregions); therefore, the sum of the left and right volumes were used for these analyses.

## Analysis of periaqueductal grey volume

The volume of the total PAG was estimated as described above. The first rostrocaudal appearance of the nucleus of Darkschewitsch (Bregma −2.80 mm) was chosen as the sample starting point and the termination of the PAG was the end point (Bregma −5.20 mm). The whole PAG was imaged and analyzed. For each brain, 12 to 16 evenly-spaced serial sections with a randomized start were studied (periodicity of every third slice; section increment of 0.12 mm). The perimeter of the PAG was digitally outlined in each section (4× magnification, N.A. 0.1) and volume was calculated. One GF mouse was excluded from PAG volumetric analysis due to damaged tissue.

# Dendritic morphology and spine density analysis

## Golgi-Cox staining procedure

Animals were terminally anesthetized with sodium pentobarbital and perfused transcardially with 0.9% saline. Golgi-cox staining was performed on the whole brain using a commercially available staining kit (Bioenno Tech, Irvine, CA). Brains were coronally sectioned at 150 µm using a vibrotome and mounted on gelatin-coated slides. Slides were lightly stained with thionin to visualize mPFC cytoarchitecture. All slides were coded to obscure the experimental group of each animal until statistical analysis.

## Analysis of dendritic morphology and spine density

The analysis of ACC pyramidal neurons was restricted to those located in cortical layers II and III. To be included in the analysis, neurons had to fulfil the following criteria (*Vyas et al., 2002*): (1) the absence of prematurely truncated dendrites, (2) dark and uniform dendritic staining, and (3) relatively isolated from neighboring neurons. Three pyramidal neurons were reconstructed in each hemisphere of each animal.

For the dendritic morphology analysis, the neurons were imaged using an Olympus AX70 Provis brightfield microscope with an Olympus DP50 camera (Mason, Ireland). Images were taken at 40× magnification (N.A. 1.0) at 1 or 2 µm intervals throughout the entire section. Neurons were reconstructed manually from color-inverted stacks using the Neurofilament tool in Imaris (Bitplane, Switzerland). Total dendritic length and branching were calculated by the software. Sholl analysis, the measurement of dendritic complexity as a function of radial distance from the soma, was performed on 2D reconstructions using the plug-in for FIJI (*Schindelin et al., 2012*; *Ferreira et al., 2014*). The radius step size was 20 µm.

Spine density and subtype analyses were performed on all ACC pyramidal neurons which were reconstructed to characterize dendritic arborization. Dendritic segments (branch order ≥2) which had consistent and dark impregnation were imaged at a magnification of 100× (N.A. 1.4). From each neuron, two to three apical and basilar dendritic segments of approximately 20 µm were analyzed. Spines were classified into subtypes using RECONSTRUCT image analysis software (*Risher et al., 2014*). Briefly, the length and width of each spine was measured and, based on these measurements, spines were classified as thin (length to width ratio >1), stubby (length to width ratio ≤1), mushroom (width value >0.6 µm), and filopodia (length value >2 µm). Values were calculated by averaging the spine data per neuron, and then once again for each animal. The data set was comprised of a total of ~7100 dendritic spines from 60 neurons.

Animal means were used for all analyses of dendritic morphology and spine density. When statistical significance was achieved, percentage changes were calculated with respect to control values.

# Statistical analysis

The sample size was determined by a power calculation and aimed at detecting differences between groups at the 0.05 level (*Lomasney et al., 2014*; *Matsuzaki et al., 2004*). Data are expressed as means + or ±1 SEM. The unpaired Student's t-test (α = 0.05) was used to compare two independent groups (CC vs GF; right vs left hemisphere). Comparisons of more than two groups were performed by one-way ANOVA. Group differences in the CRD and Sholl analyses were tested for significance

with a two-way mixed design ANOVA, with pressure and radial distance from the soma as within group factors. Post-hoc comparisons were made using a Bonferroni's correction (p<0.05). The Grubbs method (*Grubbs, 1950*) was used to test for outliers for the CRD and spinal cord analyses.

## Acknowledgements

The authors are funded by Science Foundation Ireland (SFI), through the Irish Government's National Development Plan in the form of a centre grant (APC Microbiome Institute Grant Number SFI/12/RC/2273), and through EU GRANT 613979 (MYNEWGUT FP7-KBBE-2013–7; JFC, TGD). The Centre has conducted studies in collaboration with several companies including Mead Johnson, Suntory Wellness, Cremo, Alimentary Health, 4D Pharma and Nutricia. The help of Ms. Suzanne Crotty, Ms. Tara Foley, Mr. Alan Hoban, Ms. Frances O'Brien, Mr. Patrick Fitzgerald and Dr. Roman Stilling is gratefully acknowledged.

## Additional information

### Funding

| Funder | Grant reference number | Author |
|---|---|---|
| Science Foundation Ireland | SFI/12/RC/2273 | Fergus Shanahan Timothy G Dinan John F Cryan |
| European Commission | EU GRANT 613979 (MYNEWGUT FP7-KBBE-2013-7 | Timothy G Dinan John F Cryan |

The funders had no role in study design, data collection and interpretation, or the decision to submit the work for publication.

### Author contributions

PL, Conceptualization, Data curation, Formal analysis, Investigation, Methodology, Writing—original draft, Writing—review and editing; MT, Conceptualization, Data curation, Formal analysis, Investigation, Methodology, Writing—original draft, Project administration, Writing—review and editing; MV, Formal analysis, Investigation, Methodology, Writing—review and editing; FS, Formal analysis, Writing—original draft, Project administration, Writing—review and editing; GC, Resources, Funding acquisition, Writing—review and editing; SO, Resources, Data curation, Writing—review and editing; TGD, Supervision, Funding acquisition, Project administration, Writing—review and editing; JFC, Conceptualization, Supervision, Funding acquisition, Writing—original draft, Project administration, Writing—review and editing

### Author ORCIDs

Pauline Luczynski, http://orcid.org/0000-0001-6511-2271
Monica Tramullas, http://orcid.org/0000-0002-2447-8928
John F Cryan, http://orcid.org/0000-0001-5887-2723

### Ethics

Animal experimentation: All experiments were performed in accordance with the guidelines of European Directive 86/609/EEC and the Recommendation 2007/526/65/EC and were approved by the Animal Experimentation Ethics Committee of University College Cork

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
