## [Decision Letter]

Thank you for submitting your article "Microbiota regulates visceral pain in the mouse" for consideration by *eLife*. Your article has been favorably evaluated by a Senior Editor and three reviewers, one of whom, Peggy Mason (Reviewer #1), is a member of our Board of Reviewing Editors. The following individual involved in review of your submission has agreed to reveal their identity: Jack Gilbert (Reviewer #2).

The reviewers have discussed the reviews with one another and the Reviewing Editor has drafted this decision to help you prepare a revised submission. We hope you will be able to submit the revised version within two months.

This is a simple but important study that appears to show a heightened sensitivity to colorectal dissension in GF mice which is reversed when the mice are exposed to control mice feces as adults. There are a number of molecular accompaniments observed in spinal Toll receptors and IL expression and also anatomical correlates in the ACC of the PFC and in the PAG. This is an interesting result suggesting that the microbiome has protective effects on visceral somatosensory pathways.

There are a number of methodological but serious concerns. Simply expanding the Methods section would be helpful. *eLife* is an online journal and thus space is not an issue. Please do not refer to the reader to another article for methodological details. Give the reader everything they need to understand what was done in the manuscript.

Beyond methodological details, the number of animals and cohorts used was unclear to all reviewers. It would appear that the same animals used in the CC vs GF comparison as in the CC vs GF vs GF-exposed comparison. Using the same data in 2 figures gives the false impression of more experiments having been done than were done.

The reviewers' ultimate evaluation of this manuscript will depend on their assessment of the missing methodological details and thus this request for a revision is not a pro forma antecedent to ultimate acceptance of the manuscript.

*Reviewer #1:*

This study demonstrates a heightened sensitivity to colorectal dissension in GF mice which is reversed when the mice are exposed to control mice feces as adults. There are a number of molecular accompaniments observed in spinal Toll receptors and IL expression and also anatomical correlates in the ACC of the PFC and in the PAG. This is an interesting result suggesting that the microbiome has protective effects on visceral somatosensory pathways. Yet a number of concerns need to be addressed.

Are the same animals used in the CC vs. GF comparison as in the CC vs. GF vs. GF-exposed comparison? It looks that at least some animals did double duty although the n-s don't match up exactly so this does not appear to be reuse of the entire cohort. If the same animals are used, then put them all into one figure (i.e. combine 1 and 5, 2 and 6). Otherwise it would appear that more animals were studied than were.

The CC group in Figure 1 and Figure 5 appear to be the same. This would work if the group in Figure 5 were treated as a control group (with all of the accompaniments of exposure but without the actual exposure). But without this information, it is impossible to say if this is an appropriate control group.

It is unclear why the authors jumped to the spinal cord and pfc. Isn't the simplest explanation for the finding that there is a difference in the colon itself? Why was the periphery not examined? The authors make clear that changes happen in the spinal cord and PFC but are these changes primary or secondary to changes in the periphery? Why study II/III ACC pyramidal neurons in particular?

Why is the number of animals within Figures in one group not constant? For example in Figure 1, the number varies by one. Did some animals not go through the whole protocol? This is surprising given that the protocol is straightforward and relatively short.

Reviewer #2:

The authors have shown that by germ free mice demonstrate a difference in neruo- and immune properties, which are mostly restored following colonization with a conventionally colonized microbiota. The story is a straight comparison and intervention without any specific attempt to determine which microbes are responsible for the observed activity.

I don't want to recommend any more experiments as I believe this work, while basic and yet comprehensive in its assessment of the host variables, easily stands alone. However, it would be interesting to look at the effect of microbial supernatant from a stool community, e.g. is it the cell surface proteins on intact bacteria or their metabolites and other products?

Overall, this is confirmatory work, based on expectation, but well performed (as far as I can tell – I am not a neuroanatomist, so cannot comment on that work).

*Reviewer #3:*

This report examines the role of the intestinal microbiota on visceral pain sensitivity. Using several different measures at multiple sites in the CNS, the development of visceral nociceptive processing in germfree mice is examined. The effects of transplanting normal mouse microbiota to the germfree mice on visceral pain, spinal cord TLR and cytokine expression are examined. Overall, it is an interesting and well written paper. However, there are some major issues that should be addressed. Of primary concern is that the experiments were conducted in male mice. The literature supports greater visceral sensitivity in women and female animals, suggesting females are a better model system to study. Either include experiments demonstrating no sex differences or at a minimum, explain why only males were used. What is the advantage?

Second, include some methods to explain how the fecal transplant was done. There is no information about age, duration, amount, how much transplantation is enough, how long after transplantation experiments were done.

Third, how does the transplantation affect the morphological measures shown in Figure 3 and 4. The morphological data are very interesting, but not knowing how the transplanted microbiota affect these measures, do they normalize cortical volume, makes interpretation difficult.

Fourth, perhaps there should be some discussion indicating other effects of lack of gut microbiota during development that might contribute to pain hypersensitivity.

Finally, the end of the subsection “Spinal cord signaling of visceral pain” suggests no change in TRPV1 in the GF spinal cord. This could imply that colonic nociceptor function is normal. If this is the case, it is not clear what signals the changes in spinal cord cytokine and TLRs that contribute to visceral pain. This should be further elucidated in the Discussion.

---

## [Author Response]

*Reviewer #1:*

*This study demonstrates a heightened sensitivity to colorectal dissension in GF mice which is reversed when the mice are exposed to control mice feces as adults. There are a number of molecular accompaniments observed in spinal Toll receptors and IL expression and also anatomical correlates in the ACC of the PFC and in the PAG. This is an interesting result suggesting that the microbiome has protective effects on visceral somatosensory pathways. Yet a number of concerns need to be addressed.*

*Are the same animals used in the CC vs. GF comparison as in the CC vs. GF vs. GF-exposed comparison? It looks that at least some animals did double duty although the n-s don't match up exactly so this does not appear to be reuse of the entire cohort. If the same animals are used, then put them all into one figure (i.e. combine 1 and 5, 2 and 6). Otherwise it would appear that more animals were studied than were.*

No, different animals were used in the CC vs. GF comparison and the CC vs GF vs. GFC comparison. We have outlined the cohorts of animals used in this study below and in the Methods (subsection “Animals”, second paragraph) and Results (subsection “Animals”).

Four cohorts of animals were used to perform the experiments in this paper:

1^st^ cohort (CC=10, GF=9; Figure 1): Animals underwent CRD and then were euthanized. No samples were collected.

2^nd^ cohort (CC n=9-10 and GF n=9-10; Figure 2): Animals were sacrificed without anesthesia and spinal cord tissue was collected to perform gene expression and cytokine analyses. These mice were not exposed to CRD.

3^rd^ cohort (CC=20, GF=20, GFC=10; Figure 5 and Figure 6): After data analysis, we decided that it would be of interest to determine if bacterial colonization could rescue the visceral hypersensitivity noted in GF mice. We repeated the CRD and collected the spinal cord tissue from the CC vs. GF vs. GFC cohort. Due to the limited number of GFC mice available, most GFC animals used in the spinal cord analysis had been exposed CRD. However, we verified that there was no statistical difference between those animals that underwent CRD and those that did not in the CC and GF groups. Please see the table below for a breakdown of the total animals in the 3^rd^ cohort.

Cohort 3CCGFCGFCRD1089Figure 5No CRD10121SC Tissue (All animals)202010Figure 6

4^th^ cohort (CC = 17, GF= 18; Figure 3, Figure 4, and Figure 4—figure supplement 1): All animals were euthanized without undergoing any procedures. Brain tissue from 10 animals from each group was processed for Golgi-Cox staining and the remainder of the brains were processed as per the stereology protocol.

*The CC group in Figure 1 and Figure 5 appear to be the same. This would work if the group in Figure 5 were treated as a control group (with all of the accompaniments of exposure but without the actual exposure). But without this information, it is impossible to say if this is an appropriate control group.*

The CC group in Figure 1 and Figure 5 are not the same cohort of animals. Although the CRD protocol is reliable, comparing data from GFC animals to the existing CC vs. GF data would have been methodologically unsound. To clarify this point, we now describe the cohorts of animals used in each experiment in the Materials and methods (subsection “Animals”, second paragraph) and the Results (subsection “Animals”).

*It is unclear why the authors jumped to the spinal cord and pfc. Isn't the simplest explanation for the finding that there is a difference in the colon itself? Why was the periphery not examined? The authors make clear that changes happen in the spinal cord and PFC but are these changes primary or secondary to changes in the periphery? Why study II/III ACC pyramidal neurons in particular?*

Our group’s primary focus is the interplay between the gut and the brain. Therefore, when we discovered that GF mice had heightened visceral pain sensitivity, we hypothesized this could be due to altered processing of visceral pain stimuli at the level of the central nervous system (CNS). We chose to study gene/protein expression and morphology in three key regions in the pain processing pathway: the spinal cord, periaqueductal grey, and the anterior cingulate cortex (ACC). In particular, we studied layer II/III pyramidal neurons in the ACC because this neuronal population has been shown to undergo dendritic remodelling and changes in signalling in a rodent model of chronic neuropathic pain (Metz et al., 2009).

The reviewer raises the important point that the visceral hypersensitivity in GF mice could be a result of changes to the colon itself. Indeed, GF mice have abnormal gut function and structure as well as metabolic and digestive functions (see Luczynski et al., 2016 for a review on the GF mouse model). Moreover, enteric nervous system structure and signalling is also altered in these animals (Collins et al., 2014; McVey Neufeld et al., 2013). We chose not to focus on the periphery, as it has been extensively studied by experts in gastrointestinal research. Instead, we focused on pain-relevant brain areas which have not yet been studied in GF mice. The exact mechanism by which the microbiota is influencing TLR signalling, cytokine levels, and neuronal morphology at the level of the CNS remains to determined. However, multiple primary and secondary pathways have been proposed to exist through which the gut can modulate the brain. These include endocrine (cortisol), immune (cytokines), neural (vagus and enteric nervous system), and short chain fatty acids (Cryan & Dinan, 2012). We have now added a paragraph in the Discussion addressing how the gut microbiota may be influencing and communicating with the CNS (subsection “The microbiota and visceral pain”, last paragraph).

*Why is the number of animals within Figures in one group not constant? For example in Figure 1, the number varies by one. Did some animals not go through the whole protocol? This is surprising given that the protocol is straightforward and relatively short.*

We thank the reviewer for pointing this out – In Figure 1 the varying sample size reported was an error and has now been corrected. All animals completed the protocol.

For the CRD and spinal cord analyses, the Grubbs test was used to test for outliers (see Materials and methods subsection “Statistical analysis”). In Figure 2 and Figure 6, outliers were removed for some genes studied, which explains the variation in sample size.

*Reviewer #3:*

*This report examines the role of the intestinal microbiota on visceral pain sensitivity. Using several different measures at multiple sites in the CNS, the development of visceral nociceptive processing in germfree mice is examined. The effects of transplanting normal mouse microbiota to the germfree mice on visceral pain, spinal cord TLR and cytokine expression are examined. Overall, it is an interesting and well written paper. However, there are some major issues that should be addressed. Of primary concern is that the experiments were conducted in male mice. The literature supports greater visceral sensitivity in women and female animals, suggesting females are a better model system to study. Either include experiments demonstrating no sex differences or at a minimum, explain why only males were used. What is the advantage?*

We agree with the reviewer that in the clinical setting there is a higher percentage of females reporting exaggerated visceral sensitivity. Also, some preclinical studies indicate that female rodents are more sensitive to visceral pain. These differences are often related to cyclical changes in sex hormones. This cannot explain differences that clearly also exist in the male populations with some men displaying heightened visceral pain compared to others. In these set of experiments, we aimed to assess other players, namely the gut microbiota, in determining visceral sensitivity per se. Moreover, we find across a number of different domains that the effects of the microbiome on brain and behavioural are much more robust in males than females (e.g. Clarke et al., Mol Psychiat 2013; Desbonnet et al., Brain Behav Immun 2015; Hoban et al. Trans Psychiat. 2016). We now state this clearly in the manuscript and also state that future analysis of visceral sensitivity in female animals is indeed worthy. We are quite interested in sex differences and visceral pain (e.g. Sajjad et al. Neuroscience 2016; Moloney et al., Biol Sex Differ. 2016) and have started to embark on studies investigating the interactions between sex hormones and microbiome in shaping behaviour.

*Second, include some methods to explain how the fecal transplant was done. There is no information about age, duration, amount, how much transplantation is enough, how long after transplantation experiments were done.*

We did not carry out fecal transplantation per se but rather allowed the bacteria in the conventional animal room to colonise the GF mice. We have now expanded the Materials and methods to describe in detail how GF mice were colonized with bacteria:

“A group of GF mice were colonized (GFC) on postnatal day 21 with microbiota obtained from CC mice. Briefly, GF mice were removed from the GF facility and allowed to grow to adulthood in the conventional animal facility in cages with bedding and fecal matter from CC mice, a method which has previously been shown to effectively restore a normal microbiota (Clark et al., 2013; Desbonnet et al., 2014; O’Connell Motherway et al., 2011).”

*Third, how does the transplantation affect the morphological measures shown in Figure 3 and Figure 4. The morphological data are very interesting, but not knowing how the transplanted microbiota affect these measures, do they normalize cortical volume, makes interpretation difficult.*

For the fourth cohort, we received only 18 GF mice from the GF Unit. We made the decision to perform both experiments of brain volume and dendritic morphology analyses instead of colonizing half of the GF animals and performing only one experiment. We agree that the colonization experiment would be interesting; however numerous postnatal colonization experiments have been performed in GF mice, with certain behavioral and physiological changes proving reversible (Clarke et al., 2013; Sudo et al., 2004; Heijtz et al., 2011; Hoban et al., 2016) and others irreversible (Desbonnet et al., 2014; Stilling et al., 2015; Hoban et al., 2016). With such limited availability of GF mice, we wanted to maximize the amount of pertinent information we could glean from the animals we had. We now state that the addition of colonization experiments would be of interest in future studies (subsection “Conclusions”).

*Fourth, perhaps there should be some discussion indicating other effects of lack of gut microbiota during development that might contribute to pain hypersensitivity.*

We have now added a paragraph in the Discussion indicating how other effects of lack of gut microbiota could contribute to visceral hypersensitivity in GF mice:

“Although it is now clear that the gut microbiota plays a role in visceral nociception, the exact mechanism by which the microbiota exerts its influence on the CNS remains unknown. […] Moving forward, it would be interesting to determine if is it the cell surface proteins on intact bacteria or their metabolites and other products that are involved in changes along the brain-gut axis.”

*Finally, the end of the subsection “Spinal cord signaling of visceral pain” suggests no change in TRPV1 in the GF spinal cord. This could imply that colonic nociceptor function is normal. If this is the case, it is not clear what signals the changes in spinal cord cytokine and TLRs that contribute to visceral pain. This should be further elucidated in the Discussion.*

We have now included a more detailed analysis of our TRPV1 results in the GF spinal cord in the Discussion:

“TRPV1 receptors are broadly expressed in the gastrointestinal tract and many areas of the CNS, including the spinal cord and is well-recognized as a transducer of noxious stimuli (Nagy et al. 2004). […] Moreover, GF are known to have an altered immune system and hence changes in spinal cord immune players fits well with the literature.”